

# Temporal shifts in endophyte bacterial community composition of sessile oak (*Quercus petraea*) are linked to foliar nitrogen, stomatal length, and herbivory

Luigimaria Borruso[*], Camilla Wellstein[*], Alessia Bani,
Sara Casagrande Bacchiocchi, Ania Margoni, Rita Tonin, Stefan Zerbe and
Lorenzo Brusetti

Faculty of Science and Technology, Free University of Bozen/Bolzano, Bolzano, Italy
[*] These authors contributed equally to this work.

## ABSTRACT

We studied the relationship between plant functional foliar traits and the endophytic bacterial communities associated in trees, taking the example of sessile oak (*Quercus petraea* (Matt.) Liebl). Forty-five samples with replicates of eight leaves per sample were collected in spring, summer and autumn. Bacterial community diversity was analyzed via Automated Ribosomal Intergenic Spacer Analysis (ARISA). The leaf traits specific leaf area, level of herbivory, stomatal number, stomatal length, carbon and nitrogen concentration were measured for the leaves of each sample. For statistical analysis, linear mixed effect models, the Canonical Correlation Analysis (CCA) and Non-Parametric Multivariate Analysis of Variance (NPMANOVA) were applied. Herbivory, nitrogen and carbon concentration were significantly different in autumn compared to spring and summer ($p$ value $< 0.05$), while stomatal length was differentiated between spring and the other two seasons ($p$ value $< 0.01$). The seasonal differentiation of the bacterial community structure was explained by the first and second axes (29.7% and 25.3%, respectively) in the CCA. The bacterial community structure significantly correlated with herbivory, nitrogen concentration and stomatal length. We conclude that herbivory, nitrogen content, and size of stomatal aperture at the leaf level are important for endophyte colonization in oaks growth in alpine forest environments.

# INTRODUCTION

One of the major interfaces of biological interaction is between microbiota and plants. While many integrative studies exist, regarding description of bacterial taxa related to host plants and linking bacterial and plant communities across different spatial scales (e.g., symbiosis), temporal scales remain less unexplored. Plant functional traits are increasingly used in ecological research and are a promising avenue to link plant characteristics to environmental factor in interdisciplinary researches (*Cornwell et al., 2008*; *Wellstein et al., 2011*). Leaf

Corresponding author
Lorenzo Brusetti,
lorenzo.brusetti@unibz.it

environment is characterized by foliar functional traits that are hypothesized to affect the hosted microbiome. In the context of the leaf environment of deciduous trees and inhabiting endophytic bacteria, intra-annual dynamics are of special interest.

Endophytic bacteria are ubiquitous inhabitants that colonize the inner parts of most terrestrial plant species beyond the epidermal cell layers (*Lodewyckx et al., 2015*; *Santoyo et al., 2016*). Inside the plant, there are diverse ecological niches in which endophytic bacteria can survive and grow, i.e., within cells, in the intercellular space and in the vascular systems (*Jacobs, Bugbee & Gabrielson, 1985*; *Bell et al., 1995*). Endophytic bacteria are very important to the host plant as they can contribute to the maintenance of its growth and health by, e.g., promoting nutrient acquisition and defense against pathogens (*Hirano et al., 1982*; *Afzal, Khan & Sessitsch, 2014*). This is particularly true for long-lived plant species such as trees and consequently they could affects forest ecosystems (*Griffin & Carson, 2015*; *Griffin et al., 2016*; *Griffin et al., 2017*; *Tashi-Oshnoei, Harighi & Abdollahzadeh, 2017*). Endophytic bacteria have often long-term ecological interactions with the host plants including symbiosis, mutualism and commensalism. They can either be obligate or facultative endophytes. Obligate endophytes are strictly associated with the host plant and they are eventually transferred vertically through plant generations (*Santoyo et al., 2016*). Facultative endophytes originate from the surrounding environment and they are often included within epidermal cell layers (*Hardoim, Van Overbeek & Van Elsas, 2008*). More complicate is the definition of endophytic pathogens, since historically endophytes have been defined as non-harmful microorganisms (*Hallmann et al., 1997*). Recently the advances of molecular microbiology have shown the complex dynamics of pathogenesis possibly related to the physiologic behavior of entire microbial communities, rather than of a single strain (*Fürnkranz et al., 2012*; *Erlacher et al., 2014*). In this respect, harmful and beneficial endophytes could have in common several mechanisms to colonize and diffuse into plant tissues (*Berg, Eberl & Hartmann, 2005*).

Different plant organs have diverse ways of colonization. Free-living soil microorganisms colonize roots (*Bulgarelli et al., 2012*; *Edwards et al., 2015*). Leaf endophytic bacteria, especially in case of tall trees, can be acquired from the leaf surface via stomata that represent apertures in the foliar tissue connected to the intercellular space (*Ou et al., 2014*; *Carrell, Carper & Frank, 2016*; *Griffin & Carson, 2015*). It has been hypothesized that leaves are a suitable surface for exchange with bacteria inhabiting the atmosphere (*Bowers et al., 2009*). Microbial communities inhabiting leaves, including endophytic bacteria, appear to be rather specialized, given that they share less than 1% of the bacterial species with soil (*Kim et al., 2012*).

While numerous publications are focused on leaf epiphytes (*Hirano et al., 1982*; *Balint-Kurti et al., 2010*; *Lopez-Velasco et al., 2011*), leaf endophytic bacteria remain largely unexplored. Moreover, the possible role of plant functional traits for bacterial community dynamics represents a research gap. In detail, there are a few studies regarding the temporal dynamic and the environmental factors driving the endophytic bacterial communities associated with forest tree species. Previous works revealed that endophytes are subject to leaf age and leaf developmental stage in grapevine and in elm (*Mocali et al., 2003*; *Bulgari et al., 2014*). However, the potential influencers of the endophytic bacteria composition
behind leaf aging are still not well understood, especially in forest plants. To the best of our knowledge, no papers have been published so far about a possible linkage between endophytic community assemblages, seasonality and leaf plant traits. Since it well known that stomata represents the main door for bacterial leaf colonization (*Underwood, Melotto & He, 2007*), it is reasonable hypothesize that any changes of stomata morphology due to leaf ageing could provoke consequences on the final endophytic community composition. This could reflect the previous results of *Mocali et al. (2003)* and of *Bulgari et al. (2014)*.

In our study, we aimed to test: (i) if there is a temporal gradient associating leaf aging with bacterial turnover and (ii) if foliar plant traits are linked to bacterial community dynamics across time. For this reason, we tested two hypotheses: (i) seasonality affects endophytic bacterial community structure due to changes in foliar chemical composition, and (ii) there is a strict link between endophytic bacterial community structure and foliar traits because some traits, such as stomatal length, could favor the entering of bacterial cells into leaves. To assess the validity of our hypotheses, we investigated a sessile oak forest located in the on Alps in the Northern Italy through an entire growing season.

## MATERIALS AND METHODS

### Study site and sampling

The study area is located in the Monticolo nature reserve on the hillslopes of the Mitterberg at 550 m a.s.l. in South Tyrol, Italy. The selected study site is representative of the present oak forest, dominated by sessile oak (*Quercus petraea* (Matt.) Liebl.) with few specimens of Scots pine (*Pinus sylvestris* L.) in the tree layer as well as of Sweet chestnut (*Castanea sativa* Mill.) and Manna ash (*Fraxinus ornus* L.) in the understory. The forest grows on acidic shallow soil above porphyry bedrock on a west-south-west oriented slope. We selected five individuals of sessile oak within the study site, i.e., a circular plot of 15 m radius (706 m$^2$) representing relatively homogeneous site conditions within the slope. From each tree, we selected three branches taking eight leaves from the same branch, which were used to assess the endophytic bacteria as well as the foliar functional traits and herbivory. From a previous study, it was clear which types of herbivory were generally present in the study species (*Q. petraea*) in this forest. Based on *Labandeira et al. (2007)* seven types of herbivory were found across oaks of the forest. In the present study, we considered only the most frequent herbivory types that would not affect the functional traits' analysis, i.e., margin and hole feeding and sucking *Labandeira et al. (2007)*.

Branches were chosen to have the maximum distance between them, i.e., an angle of 120 °C between two adjacent branches. In detail, we used three leaves for the functional traits and herbivory measurements and five leaves for the determination of the endophytic bacterial microbial community. We sampled three subsequent seasons in the year 2014, i.e., spring (June 5th), summer (August 25th) and autumn (October 20th). A total of 120 leaves per season was collected, 45 leaves were used for the analysis of functional traits while 75 for the analysis of endophytes.

We measured six functional traits related to important plant functions, i.e., specific leaf area (SLA), leaf nitrogen content (N), leaf carbon content (C), C:N ratio, stomatal number (STNR) and stomatal length (SL). SL is a measure for the size of stomata (*Taiz & Zeiger, 2006*). For each season, we determined the SLA of the leaves following standard protocols (*Pérez-Harguindeguy et al., 2013*). For each leaf, the area was measured the sampling day using a scanner (CanoScan Lide, Canon, Cernusco sul Naviglio, Italy). Subsequently, leaves were oven dried at 70 °C for 72 h to obtain their dry weight and the SLA, measured in $mm^2$ $mg^{-1}$, was calculated (*Pérez-Harguindeguy et al., 2013*). N and C were determined using an elemental analyzer (Flash 2000 Organic Elemental Analyzer, Thermo Scientific, Milan, Italy) pooling together the three leaves of each branch.

To measure stomatal characteristics, we applied the clear nail polish method described by *Hilu & Randall (1984)* obtaining epidermal impressions of the abaxial surface of each leaves that were examined under an optical microscope (Leica DMLS, Leica Biosystems, Nussloch, Germany) connected to a digital camera. The images were analyzed through the image processing software DeltaPix InSight, (DeltaPix, Smorum, Denmark). The stomata were counted on three fields of view per leaf on a standard counting area at $400\times$ magnification to determine the stomatal density as number of stomata (STNR) for each standard counting area. On each counting area, the length of the guard cells of stomata (SL) was measured for 15 randomly selected stomata.

For the measurement of herbivory, the percentage of consumed leaf area was measured scanning the leaf surface (CanoScan LiDE 120; Canon, Tokyo, Japan) and calculating the lost area using free software ImageJ (https://imagej.net). Because the great majority of the leaves presented a mixture of the investigated herbivory mediated damage types, for each leaf we quantified the total surface damage caused by the summed herbivory damage types.

## DNA extraction and Automated Ribosomal Intergenic Spacer Analysis (ARISA)

Leaves used for microbiological analysis were processed within 4 h as follows (K Hrynkiewicz, pers. comm., 2015): five leaves for each replicate were disinfected with 70% ethanol twice each for 3 min. Leaves were then washed with sodium hypochlorite (1.5%) and TWEEN® 20 for ten minutes, three rinses in sterile, distilled shaking water. Disinfected leaves were grinded to a fine powder under liquid nitrogen using a sterile mortar and pestle. The disinfected samples were stored at −20 °C. Triplicates of the water used in the last rise were used as negative for PCR amplification and plated on a LB and TSA medium to verify the disinfection protocol. Absence of PCR amplification products was observed. Furthermore, absence of bacterial colonies was observed in all the plates after 10 days of incubation at 30 °C.

DNA was extracted using the Qiagen DNeasy PowerPlant Pro Kit (Qiagen, Milan, Italy) accordingly to the user's manual. Extracted DNA was stored at −80 °C. The quality and the size of the soil DNA were checked by electrophoresis on 1.2% agarose gel with a marker (Eurogentec Smart Ladder, Belgio). The absorbance (260 nm) of two µl of DNA was used to evaluate the concentration of DNA by NanoVue Spectrophotometer (GE Healthcare, Little Chalfont, UK).

The 16S-23S rRNA Internal Transcribed Spacer (ITS)-PCR was performed using the primers ITSF and ITSReub labeled with 6-FAM according to the chemical and thermal amplification protocol of *Cardinale et al. (2004)*. Capillary electrophoresis was done by STAB Vida Lda. (Caparica, Portugal). Data were investigated via Peak Scanner Software 1.0 (Applied Biosystems, Monza, Italy) and the downstream matrix was normalized and analyzed according to *Borruso, Zerbe & Brusetti (2015)*.

### Data analysis

PAST software (*Hammer, Harper & Ryan, 2001*) was used for the statistical analysis. ANOVA was used to test for differences in the endophytic bacterial richness between seasons. Canonical correspondence analysis (CCA) of the endophytic microbial community structure in dependence of functional leaf traits and season was performed. Non-Parametric Multivariate Analysis of Variance (NPMANOVA) with Bonferroni corrected $p$-value was applied to investigate differences among the endophytic bacterial communities across the three seasons using Bray-Curtis dissimilarity distance.

Linear discriminant analysis (LDA) effect size (LEfSe) algorithm was used to identify taxa preferentially abundant in each season using default parameters. Briefly, the algorithm identifies the indicator bacterial taxa specialized within the 3 seasons (*Segata et al., 2011*).

Given the nested design of the experiment, the variation of leaf functional traits across the three seasons was investigated applying linear mixed effect models in R (R Development Core Team 2014 version 3.1.2), using *nlme* package (*Pinheiro et al., 2017*). For SLA, SL, STNR, N, C, C:N and herbivory we analyzed each trait as response variable, season as fixed variable and, as random factor, we nested the branches from which we sampled the leaves in the respective trees. We log transformed the data that did not satisfied the assumption of variance normality tested with Shapiro test.

## RESULTS

### Functional leaf traits and insect-mediated damage types

Figure 1 and Table 1 shows the results of functional leaf traits investigated in this study across the three sampling seasons. Leaf N and C were significantly lower in autumn than in spring and summer ($p < 0.01$). The C:N ratio and the level of herbivory were significantly higher in autumn than in spring and summer ($p < 0.01$). SL was significantly higher in spring compared to summer and autumn ($p < 0.01$). STNR and SLA did not show significant differences across the seasons ($p > 0.05$).

### Bacterial community structure

An average of $82 \pm 15$ peaks per sample representing bacterial richness, ranging from 200 bp to 1,200 bp, were found. No significant results in terms of number of peaks across the three seasons were found (spring $82 \pm 9$; summer $86 \pm 19$ and autumn $79 \pm 14$; ANOVA $p$-value: n.s.). NPANOVA showed significant differences between the bacterial community structure of spring and of autumn ($p < 0.001$), while the bacterial community structures of summer did not cluster apart representing a bridge between the two seasons (Table 2). Scattered peaks in between the range 550–850 bp were mostly found in spring and autumn,

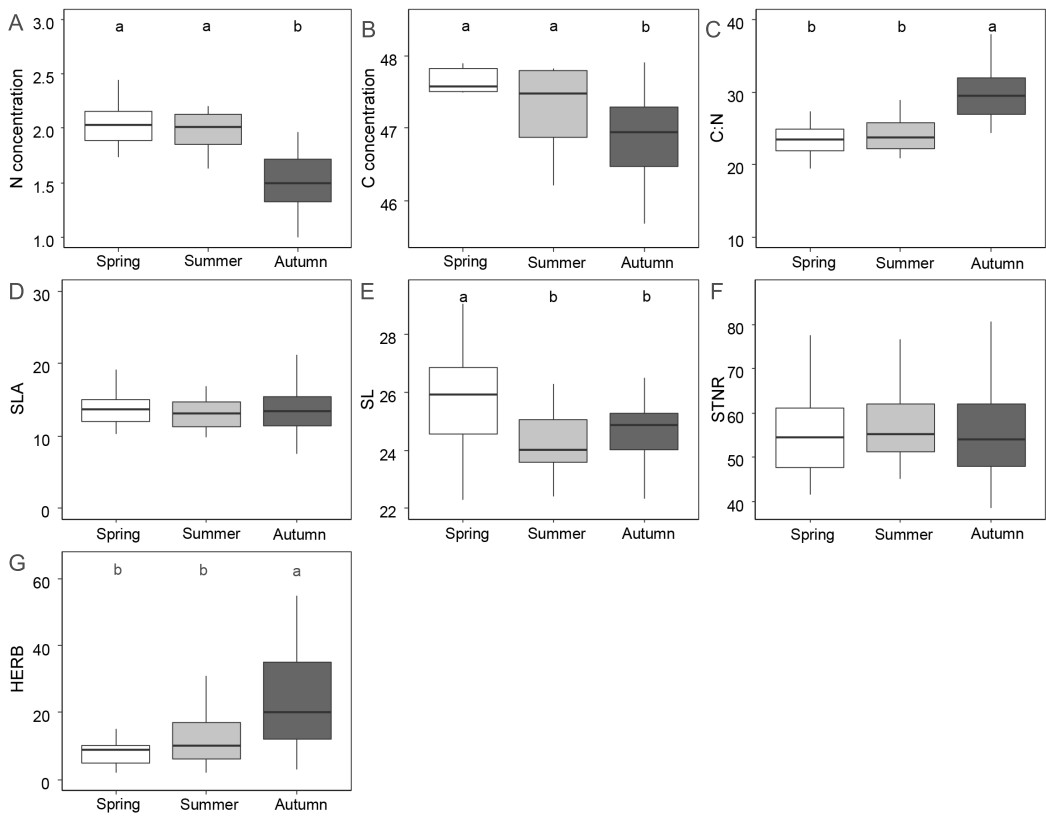

**Figure 1** **Differences of foliar traits (N, nitrogen; C, carbon; C:N, carbon/nitrogen ratio; SLA, specific leaf area; SL, stomatal length; STNR, stomatal number per reference area; HERB, level of herbivory) among three seasons (spring, summer, autumn).** Significant differences according to linear mixed effect models followed by post-hoc test are indicated by different lower case letters. Graphics without letters were not significant. Detailed results of linear mixed effect models are given in Table 3.

causing the separation between these two seasons. The LDA showed that 13 ARISA peaks were responsible of the discrimination of spring with respect to the other seasons, 14 peaks were typically discriminant for summer, and 9 for autumn. Most of those peaks were situated in the 550–880 bp slot, confirming the NPANOVA results (Table 3).

## Canonical correspondence analysis (CCA)

Canonical correspondence analysis (CCA) was used to investigate the effects of functional leaf traits and the level of herbivory on endophytic bacterial communities across the seasons. Differentiation is illustrated by the first and second axes in the CCA (29.7% and 25.3%, respectively) and the leaf features fitted into the CCA. The CCA ordination diagram (Fig. 2) revealed first, that community structure variation appeared along season (temporal sequence) and, second, the existence of relationships between plant foliar traits and endofoliar microbiota across the temporal sequence. In detail, the community variation is related mainly to the level of herbivory, N and SL, and less to STNR, C and SLA (length of vectors in the ordination diagram; Fig. 2).

**Table 1** **Results of linear mixed effect models for each trait (N, leaf nitrogen content; C, leaf carbon content; C:N, C:N ratio; SLA, specific leaf area; SL, stomatal length; STNR, stomatal number; HERB, herbivory level).** Each single trait was analyzed as response variable, season as fixed variable and branches nested in the respective trees as random variable. The basic level (intercept) corresponds to the spring season.

| Trait | Fixed effect | Value | Std.error | DF | t-value | p-value |
|---|---|---|---|---|---|---|
| N | (Intercept) | 2.04 | 0.07 | 28 | 30.67 | 0.00 |
| | Summer | −0.07 | 0.09 | 28 | −0.85 | 0.41 |
| | Autumn | −0.57 | 0.09 | 28 | −6.51 | 0.00 |
| log(C) | (Intercept) | 3.86 | 0.00 | 28 | 882.56 | 0.00 |
| | Summer | −0.01 | 0.00 | 28 | −1.68 | 0.10 |
| | Autumn | −0.02 | 0.00 | 28 | −4.90 | 0.00 |
| log(C:N) | (Intercept) | 3.16 | 0.05 | 28 | 69.09 | 0.00 |
| | Summer | 0.03 | 0.06 | 28 | 0.54 | 0.59 |
| | Autumn | 0.33 | 0.06 | 28 | 5.82 | 0.00 |
| SLA | (Intercept) | 13.98 | 0.99 | 28 | 14.05 | 0.00 |
| | Summer | −0.84 | 1.26 | 28 | −0.66 | 0.51 |
| | Autumn | 0.97 | 1.26 | 28 | 0.77 | 0.45 |
| SL | (Intercept) | 25.69 | 0.58 | 28 | 44.00 | 0.00 |
| | Summer | −1.66 | 0.31 | 28 | −5.44 | 0.00 |
| | Autumn | −1.49 | 0.31 | 28 | −4.87 | 0.00 |
| STNR | (Intercept) | 55.48 | 3.87 | 28 | 14.35 | 0.00 |
| | Summer | 2.37 | 1.72 | 28 | 1.38 | 0.18 |
| | Autumn | 0.81 | 1.72 | 28 | 0.47 | 0.64 |
| log(HERB) | (Intercept) | 2.08 | 0.13 | 28 | 16.29 | 0.00 |
| | Summer | 0.39 | 0.17 | 28 | 2.30 | 0.03 |
| | Autumn | 0.88 | 0.17 | 28 | 5.16 | 0.00 |

**Table 2** **P value results from Non-Parametric MANOVA (NPMANOVA) with Bonferroni corrected p value among endophytic bacterial communities across the three seasons (Bray-Curtis dissimilarity).**

| Seasons | Summer | Autumn |
|---|---|---|
| Spring | 0.0804 | 0.0009 |
| Summer | / | 0.2484 |

## DISCUSSION

We explored the seasonal diversity behavior via fingerprinting ARISA of the of leaf endophytic bacterial communities. ARISA is a corroborate technique used to investigate bacterial structure variations and the correlations with environmental parameters (*Esposito et al., 2013*; *Borruso, Zerbe & Brusetti, 2015*; *Pioli et al., 2018*) with a comparable robustness as Next Generation Sequencing (*Van Dorst et al., 2014*). Differently by other fingerprinting techniques such as the Length Heterogeneity-PCR or the Denaturing Gradient Gel Electrophoresis of the partial 16S rRNA genes, ARISA can investigate the bacterial community at a deeper taxonomic resolution. Actually, ARISA can reach the subspecies level (*Danovaro et al., 2006*), via detection of the length polymorphisms of the internal

**Table 3  OTU biomarkers characterizing each single season on the basis of the Linear Discriminant Analysis effect size.**

| OTU | Season | LDA-score | *p*-value |
| --- | --- | --- | --- |
| OTU620 | Autumn | 3.22 | 0.033 |
| OTU660 | Autumn | 3.51 | 0.016 |
| OTU713 | Autumn | 3.26 | 0.043 |
| OTU833 | Autumn | 2.93 | 0.043 |
| OTU582 | Autumn | 4.05 | 0.000 |
| OTU600 | Autumn | 3.24 | 0.035 |
| OTU1160 | Autumn | 2.83 | 0.039 |
| OTU640 | Autumn | 3.09 | 0.034 |
| OTU850 | Autumn | 3.06 | 0.035 |
| OTU222 | Spring | 4.38 | 0.000 |
| OTU454 | Spring | 4.00 | 0.000 |
| OTU671 | Spring | 3.07 | 0.043 |
| OTU530 | Spring | 3.15 | 0.004 |
| OTU286 | Spring | 3.18 | 0.007 |
| OTU361 | Spring | 3.90 | 0.000 |
| OTU213 | Spring | 2.95 | 0.014 |
| OTU547 | Spring | 3.65 | 0.043 |
| OTU238 | Spring | 3.68 | 0.050 |
| OTU383 | Spring | 3.23 | 0.043 |
| OTU474 | Spring | 3.68 | 0.043 |
| OTU654 | Spring | 4.36 | 0.004 |
| OTU519 | Spring | 3.79 | 0.024 |
| OTU350 | Summer | 3.08 | 0.008 |
| OTU575 | Summer | 3.27 | 0.042 |
| OTU573 | Summer | 3.54 | 0.030 |
| OTU316 | Summer | 2.98 | 0.014 |
| OTU842 | Summer | 2.67 | 0.043 |
| OTU598 | Summer | 3.13 | 0.015 |
| OTU662 | Summer | 3.16 | 0.030 |
| OTU666 | Summer | 2.87 | 0.014 |
| OTU529 | Summer | 3.17 | 0.004 |
| OTU995 | Summer | 2.91 | 0.014 |
| OTU727 | Summer | 2.79 | 0.043 |
| OTU1119 | Summer | 2.70 | 0.014 |
| OTU564 | Summer | 2.79 | 0.043 |
| OTU634 | Summer | 2.82 | 0.043 |

16S-23S ribosomal DNA spacers within the several copies of ribosomal operons in a bacterial cell (typically from 1 to 10 copies; *Gürtler, 1999*). Although the endophytic bacterial communities did not show significant differences in alpha diversity across the three seasons, their beta diversity differed mainly between spring and autumn (Table 2 and Fig. 2). These results support the idea of an intimate association between endophytes

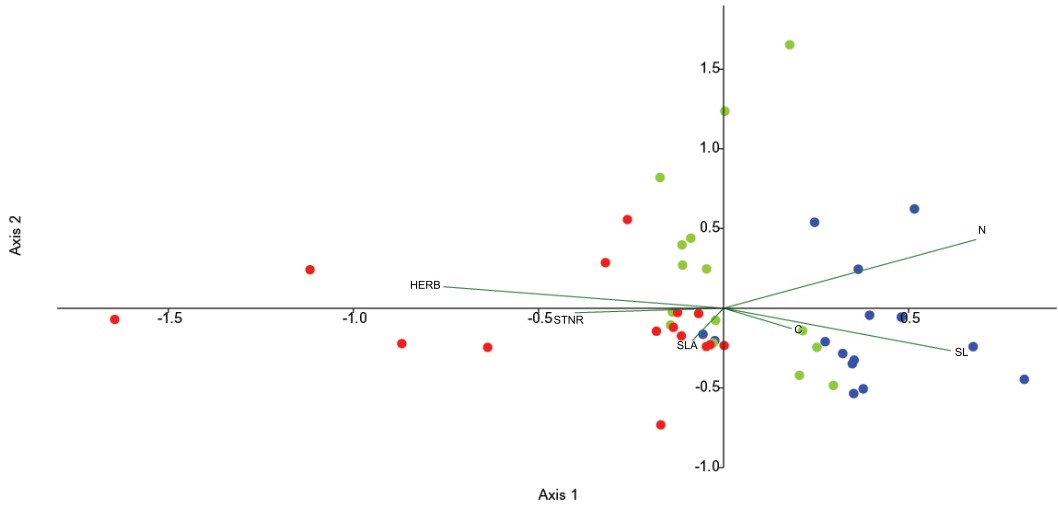

**Figure 2  CCA analysis of endophytic communities across the three seasons.** CCA analysis of endophytic communities across a temporal sequence (spring, blue dots; summer, green dots; autumn, red dots) and plant foliar traits. CCA was calculated with the following plant foliar traits: HERB, level of herbivory; STNR, number of stomata; SLA, specific leaf area; SL, length of stomata; N, leaf nitrogen content; C, leaf carbon content.

and the leaf, seen as a dynamic micro-ecosystem that selects for different specific microbial communities along time. Regarding which taxa contribute to the observed difference between seasons, even if the attribution of single ARISA peaks to specific taxa cannot be conclusive due to the real possibility that a peak could be represented by several taxa from different phyla, a putative raw taxon attribution could be done. For instance, according to some authors, many Gram negative bacterial species harbor ITS with tRNA genes (*Fisher & Triplett, 1999*; *Ranjard et al., 2001*). Their spacers usually range between 500 and 800 bp (Gürtler and Stanisich, 1996). On the other side Gram positive bacteria harbor shorter ITS, while rhizobia have very long spacers, often longer than 1,000 bp (*Gürtler & Stanisich, 1996*). The length range between 500 and 600 bp is what had been observed in our case differentiating spring to autumn. We could hypothesize that seasons have an effect on the diversity of Gram negative bacteria. A different answer of Gram negative bacteria rather than Gram positive bacteria due to different environmental pressures had already been observed in other plant-related compartments, such as rhizosphere (*Ciccazzo et al., 2014*).

Leaves have been traditionally considered as "short-lived environment", where specialized bacteria can dynamically colonize new niches and leave others according to the leaf continuous modifications over seasons (*Vorholt, 2012*). *Bulgari et al. (2014)* hypothesized that the endophytic communities in *Vitis vinifera* should remain stable across the seasons in absence of bacterial plant pathogens such as phytoplasma. However, this is in contrast with our findings and those of other researchers. Influence of the seasonality on endophytic microbial community composition associated with different tree species (i.e., *Acer negundo*, *Ulmus pumila*, and *U. parvifolia*) were also found by *Shen & Fulthorpe (2015)*. Moreover, others observed that the bacterial community composition

in the phyllosphere was primarily driven by temporal changes and community succession (*Copeland et al., 2015*). In order to shed light on the processes behind community changes, we explored the role of leaf functional traits in bacterial structure dynamics. In fact, previous studies investigated the effect of migration and community succession in the phyllosphere microbiome and suggested that colonization, persistence, and succession of the community may be key-factors driving the phyllosphere microbiome (*Redford & Fierer, 2009*; *Shade, McManus & Handelsman, 2013*; *Maignien et al., 2014*; *Copeland et al., 2015*). The relevance of environmental conditions such as temperature optima of the bacteria or the changing physiology of tree host species have been discussed for their possible effects (*Jansson & Douglas, 2007*). However, previous studies did not look deeper into possible correlations in terms of plant functional characteristics behind the observed dynamical processes of the microbiome.

Given that we found a clear endophytic seasonal variation, it is highly interesting to understand better the potential leaf-level characteristics possibly associated with bacteria community compositional variation. In fact, based on our results, we suggest that foliar characteristics related to leaf-level herbivory, nutrient contents and stomatal length aperture may affect directly the bacterial community composition over time.

As a matter of fact, herbivory could be an effective way to inoculate microbial insects symbionts, commensals and pathogens into plant tissues. It is the case, for instance, of phytoplasmas inoculated into grapevine by Hemiptera-like leafhoppers (*Gonella et al., 2008*; *Alma et al., 2018*). In our forest plots, several herbivory traces have been recorded during the experiments. In general, the majority of insect species causes single distinct damages on leaves (*Labandeira et al., 2007*). From a survey on *Quercus petraea*'s leaves in the same forest, the herbivory damage types were recorded by using three leaves per individual with a total amount of 180 plant individuals. Seven types of damages were recorded: Margin feeding (detected in the 90% of the observed leaves), surface feeding (61%), hole feeding (45%), sucking (31%), skeletonization (12%), mining (12%) and leaf rolling (9%). This number of herbivory damage types justifies the perception of a relatively high insect diversity since it has been shown that there is a quantitative relation between the richness of damage types and the insect species richness (*Carvalho et al., 2015*). Phloem-feeding insects could act as inoculating vectors of entire bacterial communities between different plant individuals, moving bacterial strains from a tree to another (*Lòpez-Fernàndez et al., 2017*). It is reasonable to hypothesize that, as insect abundance and diversity may change due to season variation (*Grimbacher et al., 2018*), also the endophytic bacterial communities potentially transmissible among tree individuals change, hence contributing to our observed results.

In addition, stomata are the major door for the leaf colonization by foliar bacterial pathogens (*Underwood, Melotto & He, 2007*; *Melotto, Underwood & He, 2008*). To counteract the entrance of potential pathogens, plants have evolved a number of mechanisms to detect and remove pathogenic bacterial cells from their tissues, regulating to some extent the access of bacteria through stomata (*Gimenez-Ibanez et al., 2017*). This is due to the ability of plants in detecting specific molecular signals such as bacterial lipopolysaccharides, flagellins or elongation factors (*Underwood, Melotto & He, 2007*).

Consequently, plants close the majority of their leaf stomata. However, bacteria may overcome this mechanism of defense, entering into the leaf intercellular spaces by taking advantage of those stomatal guard cells that are not able to react to the presence of bacterial signals (*Underwood, Melotto & He, 2007*). We observed that SL could be associated with the shaping of endophytic community structure, especially in spring (Fig. 2). This observation could mirror the entrance of specific as well as unspecific bacterial strains into leaves when the leaves were growing. Actually, *Gailing et al. (2008)* found that in *Quercus robur* the variability of stomatal number is genetically determined. Additionally, *Turner & Heichel (1977)* demonstrated for *Quercus rubra* that SL reaches it maximum before leaves developed their maximum areas in late spring. It is plausible that bacteria enter into the leaf as soon as it has flashed, and then a sort of successional dynamic is established until reaching an equilibrium once the stomata shape are fixed. The difference in SL between spring leaves from summer and autumn leaves may depend by the contraction of the leaf pool available for measurements caused by the loss of the leaves that occur both naturally during the vegetative season (*Brooke et al., 1996*) and because of the detected herbivory.

Finally, leaf nutrient contents and their changes correlate with bacteria dynamics (*Kembel et al., 2014*). It has been shown that endophytic bacterial taxa able to fix nitrogen occur in oak species (*Tashi-Oshnoei, Harighi & Abdollahzadeh, 2017*), in the wild poplar *Populus trichocarpa* leaves (*Doty et al., 2016*), and in *Pinus flexilis* needles (*Moyes et al., 2016*), helping plants establishment and growth in N-limited environment. Eventually, a drop in leaf nitrogen content could challenge bacterial shifts towards nitrogen-fixing taxa and in contrast to denitrifying bacteria.

Our results of diminishing nutrients such as C and N with leaf aging is in line with other studies (*Li et al., 2017*), while other functional traits related to the stomata, such as STNR and SLA did not vary significantly along season. Another study on leaf traits of seven different woody species grown under experimental conditions shed light on their seasonal variation (*Römermann et al., 2016*). The results of this study highlight that SLA and stomatal size were robust traits across season in terms of small intraspecific variation. In comparison, our species *Q. petraea* also has stable STNR and SLA levels. However, as explained above, the changes in SL most likely reflect changes in the leaves' pool of the forest as the leaves' pool was diminished by herbivory and browsing that led to leaf loss after the spring season. Moreover, herbivory, that increases over season due to elongated exposure time, can have a direct impact on endophytic bacteria as well as an indirect effect by influencing other leaf characteristics. For example, the open structures of the leaf, limited to the size and number of stomata in intact leaves, are largely modified by herbivory that exposes further leaf tissue. Due to the increasing rate of changes that can be assumed with aging (*Suzuki et al., 1987*; *Chavana-Bryant et al., 2017*), we expect that leaf characteristics exert a differential impact during aging on the bacterial community.

## CONCLUSIONS

Based on our findings, we suggest that herbivory, nitrogen content, and size of stomatal aperture at the leaf level are important for endophyte colonization in oak growth in alpine

forest environments. We argued that herbivory and stomata length are important doors from where bacteria enter to colonize the leaf. As possible consequence, the endophytic community assemblages switch during the progression of seasons, when the stomatal length increases during the leaf germination and elongation, and when the chemical characteristics of the leaf are different from those in autumn.

## ACKNOWLEDGEMENTS

We thank Prof. Giustino Tonon for the access to the Monticolo site area, managed under the NITROFOR project.

### Funding

This work was supported by the project "Effects of forest aerial fertilization on the microbial communities of coniferous tree phyllosphere, residuosphere and rhizosphere –MICRONITRAIR" (CUP: I52I13000180005). Partial contributions came from the project entitled "Multidisciplinary characterization of a forest-ecosystem supersite – MULTFOR" (CUP: I52I13000260005). Both projects were granted by the Free University of Bozen-Bolzano. Finally, Luigimaria Borruso received a grant from the "Transdisciplinary Environment and Health Research Network South Tyrol" (TER). There was no additional external funding received for this study. The funders had no role in study design, data collection and analysis, decision to publish, or preparation of the manuscript.

### Grant Disclosures

The following grant information was disclosed by the authors:
Effects of forest aerial fertilization on the microbial communities of coniferous tree phyllosphere, residuosphere and rhizosphere–MICRONITRAIR: I52I13000180005.
Multidisciplinary characterization of a forest-ecosystem supersite–MULTFOR: I52I13000260005.
Free University of Bozen-Bolzano.
Transdisciplinary Environment and Health Research Network South Tyrol (TER).

### Competing Interests

The authors declare there are no competing interests.

### Author Contributions

- Luigimaria Borruso and Camilla Wellstein conceived and designed the experiments, performed the experiments, analyzed the data, prepared figures and/or tables, authored or reviewed drafts of the paper, approved the final draft.
- Alessia Bani, Sara Casagrande Bacchiocchi, Ania Margoni and Stefan Zerbe performed the experiments, authored or reviewed drafts of the paper, approved the final draft.
- Rita Tonin performed the experiments, analyzed the data, approved the final draft.
- Lorenzo Brusetti conceived and designed the experiments, contributed reagents/materials/analysis tools, authored or reviewed drafts of the paper, approved the final draft.

## Data Availability

The raw data are provided in the Supplemental File.

## Supplemental Information

Supplemental information for this article can be found online at http://dx.doi.org/10.7717/peerj.5769#supplemental-information.

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
