# Peer review of "Temporal shifts in endophyte bacterial community composition of sessile oak (Quercus petraea) are linked to foliar nitrogen, stomatal length, and herbivory"

_PeerJ, doi:10.7717/peerj.5769_

## Round 0.1 · original submission · Major Revisions

Both reviewers noted that the the description of the experimental design needs more clarification, due to the high level of nestedness adopted. More details are needed on how spatial and temporal autocorrelation were addressed.

Reviewer 1 ·

Basic reporting

No comment here.

Experimental design

Why did you use three leaves for functional measurements and five for endophyte characterizations? Were the same leaves used for both functional and endophyte analysis? Were the leaf samples evaluated independently? If multiple samples were taken from branches and within trees and evaluated independently, this study is pseudoreplicated and the samples from each branch and tree should have been combined (I see that you did this properly for stomatal measurements). But I feel that this is not the case for endophyte analysis, because the authors state that 45 samples were taken from only 5 individuals. This is important to specify.

Moreover, if all five individuals occur within a 15-m radius, your conclusions might be limited and potentially confounded, because conspecifics in closely spaced habitats can share microbes, even endophytes that disperse on the leaf surface and ultimately gain access to leaf interiors (Griffin & Carson 2015).

Validity of the findings

The Results and consequently the Discussion sections are too short and do not offer much insight on detailed taxa-specific results and how these results may impact plant hosts and tree populations at larger scales. Sure, one would expect that bacterial community (richness? What about entire community structure?) changes with season, but which particular taxa drive these differences? What is known about these taxa? Are they pathogens, mutualists, are they correlated with herbivory or potentially have effects on other trophic levels and ultimately plant performance? The ecology of the results and how they are important are needed.

And what temporal-plant functional trait interactions are there? For instance, it appears that endophyte communities are particularly structured by stomatal length (SL) in spring (Fig. 2). There are many other trends to speculate and explore in the Discussion. This type of synthesis is needed and will greatly increase the quality, scope, and contribution to the field.

It was quite surprising to see that even though the authors concluded that nitrogen content and herbivory were important in structuring endophyte communities, there was little mention other than fixation of the potential physiological effects of nitrogen or herbivory. What might be happening? This should be explored further in the Discussion and possibly even mentioned in the Introduction. For a starting point, look at Dordas 2008 review from the agricultural literature about nutrients. What are the herbivores in the system, and how are ways herbivores drive endophytes? Are they vectoring bacteria among plant hosts? See recent paper by Vannette & Fukami (2017) or reviews by Biere & Bennett 2013 and Griffin & Carson 2015.

Overall, this paper is skeletal and needs a lot more meat, particularly in the Discussion. These are good results and the authors should do this paper justice by discussing some of the important implications of the findings.

Additional comments

This study aims to determine the relative importance of temporal variation and plant functional traits in structuring bacterial endophyte communities of Quercus petraea. The plant microbiome and the factors that structure these communities are timely important areas of research, particularly among forest trees. While the scope of the study and the results are interesting, there are significant issues with the experimental design, scholarship, and overall skeletal prose of the manuscript. My detailed comments are below.

Why did you use three leaves for functional measurements and five for endophyte characterizations? Were the same leaves used for both functional and endophyte analysis? Were the leaf samples evaluated independently? If multiple samples were taken from branches and within trees and evaluated independently, this study is pseudoreplicated and the samples from each branch and tree should have been combined (I see that you did this properly for stomatal measurements). But I feel that this is not the case for endophyte analysis, because the authors state that 45 samples were taken from only 5 individuals. This is important to specify.

Moreover, if all five individuals occur within a 15-m radius, your conclusions might be limited and potentially confounded, because conspecifics in closely spaced habitats can share microbes, even endophytes that disperse on the leaf surface and ultimately gain access to leaf interiors (Griffin & Carson 2015).

The Results and consequently the Discussion sections are too short and do not offer much insight on detailed taxa-specific results and how these results may impact plant hosts and tree populations at larger scales. Sure, one would expect that bacterial community (richness? What about entire community structure?) changes with season, but which particular taxa drive these differences? What is known about these taxa? Are they pathogens, mutualists, are they correlated with herbivory or potentially have effects on other trophic levels and ultimately plant performance? The ecology of the results and how they are important are needed.

And what temporal-plant functional trait interactions are there? For instance, it appears that endophyte communities are particularly structured by stomatal length (SL) in spring (Fig. 2). There are many other trends to speculate and explore in the Discussion. This type of synthesis is needed and will greatly increase the quality, scope, and contribution to the field.

It was quite surprising to see that even though the authors concluded that nitrogen content and herbivory were important in structuring endophyte communities, there was little mention other than fixation of the potential physiological effects of nitrogen or herbivory. What might be happening? This should be explored further in the Discussion and possibly even mentioned in the Introduction. For a starting point, look at Dordas 2008 review from the agricultural literature about nutrients. What are the herbivores in the system, and how are ways herbivores drive endophytes? Are they vectoring bacteria among plant hosts? See recent paper by Vannette & Fukami (2017) or reviews by Biere & Bennett 2013 and Griffin & Carson 2015.

Overall, this paper is skeletal and needs a lot more meat, particularly in the Discussion. These are good results and the authors should do this paper justice by discussing some of the important implications of the findings.

There were quite a few instances where the prose was not clear or grammatically incorrect. I suspect that there were some difficulties in a language barrier. For a future submission, I would recommend that the authors send the manuscript to a colleague whose first language is English for detailed comments. I comment on (some) instances below.

Moreover, there were quite a few instances where important papers/citations were not used, or where more relevant and recent citations should be used. This demonstrates a lack of scholarship, and makes the reader question whether the authors are up to speed on the literature. See detailed examples below.

Recommended Title: Herbivory, nitrogen, and stomatal density structure endophyte bacterial community composition of sessile oak (Quercus petranea)

Line 43: “In this sense…” What sense? Incorrectly used
Line 50: Kembel et al. 2014 did not study endophytes
Line 54: Affect not affects
Lines 54-55: cite also Griffin et al. 2016, Griffin et al. 2017
Line 65: reviewed by Griffin & Carson 2015
Line 68: Great point
Line 78: I strongly suggest briefly outlining at least two hypotheses here.
Line 91: Were branches randomly selected?
Line 93-94: I think there are some issues with the experimental design. See my comments above.
Line 96: “were” collected
Lines 108-112: Citations for stomata counting and leaf herbivory? Metrics for measuring herbivory have widely been scrutinized. What are the enemies and their impacts among this species? There is no natural history given here or even in the introduction.
Line 117: “Tween?”
Cite your protocols for surface-sterilization: Arnold & Lutzoni 2007, Griffin et al. 2016
Line 129: I do not know much about ALISA, but after a quick search it appears that ARISA can either underestimate (multiple taxa sharing an ITS length) or overestimate (more than one copy of rRNA operon encoding ITS region). At the very least, the authors should address this limitation with proper citations.
Line 206: cite Kembel et al. 2014 for phyllosphere bacteria, other citations needed
Line 207: also cite Doty et al. 2016 for N fixation in Populus
Line 220: need citation
Line 153: No information about how community structure differs, no info on particular bacterial taxa. There is no biology here.
There is no discussion because there are no exploration of particular results. What about particular taxa? What is herbivory doing? What is time doing?
Line 215: over seasons rather than “along season”
Line 217: Impacts rather than impact
Line 220: needs citation
Line 223: This single sentence conclusion section is disparate and should either be expanded or incorporated into the Discussion.

How much variation is there within trees? 5 isn’t very well replicated for a microbial study.

·

Basic reporting

Overall comments:
This paper suggests that temporal variation in bacterial assemblages is present within the leaves of sessile oak. The most interesting finding was that leaf herbivory was an important predictor of community divergence. I commend the authors on pursuing timely questions and presenting some interesting findings.

The main suggestions I have to improve the paper are:

1. The statistics employed do not appear to account for non-independence among samples imposed by temporal or spatial autocorrelation.
2. The introduction is very brief and does not give the reader much information regarding what to expect or why the study was performed. Providing more information here would make the paper read much better.

More detailed suggestions follow.

Title: The phrase “community structures” is somewhat vague. Perhaps using slightly different phrasing would be better.

Introduction
Lines: 36-37. This introductory sentence seems quite broad and somewhat inappropriate. Perhaps you could start by saying how not much is known about temporal variation in microbial assemblages (as you suggest in line 38). The appeal for interdisciplinary work seems unnecessary.

Second paragraph: no mention is made of the potentially harmful effects of endophytic bacteria. It may be worth providing a citation for the reader that shows how some bacteria likely interact antagonistically with their hosts.

Line 61: This sentence is unclear as written.
Line 69: This sentence seems to contradict itself, perhaps it would be best to rewrite.
Line 70 onwards: It may be worth setting up some expectations here for why certain plant traits may influence endophyte assemblages. For instance, perhaps SLA is important because it represents within-leaf habitat. Also, additional citations here would help many readers. Why do we think leaf age may be associated with shifts in endophyte assemblages?

Additionally, there is no information about why we should expect endophyte communities to vary with time, or the many studies that have shown temporal variation does exist for both endophytic bacteria and fungi. There probably should be a whole paragraph describing why temporal variation matters and setting up expectations.

Overall the introduction has some good background information but could be improved by better setting up expectations and justifications for your study.

Methods

Line 86: admixed does not seem like the best word here.
Line 137: Was a repeated measures approach taken when calculating the ANOVA to account for temporal autocorrelation? If not, then this should be performed.

Additionally, the statistics need to account for the highly nested nature of your data. You sampled multiple leaves from the same branch, and multiple branches from the same tree. These leaves are therefore not independent measurements and you will want to account for that statistically. A mixed effect model is an easy way to do this. Mixed effects models can be specified quickly in the R environment (see the lme4 package).

Line 140: the Bonferroni measure is very conservative. The Benjamini-Hochberg false discovery rate correction may be a better choice. It is possible that the greater sensitivity of the latter test might expand your inferences slightly.

It would be helpful to provide readers more information about the ARISA technique in the methods or introduction as I suspect many people will be unfamiliar with this method. Also the acronym is never defined.


Results

Line 156: I have used the phrase “community structure” in my own work, but have grown dissatisfied with the phrase because it is so vague. Could you perhaps say that “community composition” shifted?

Line 168: It is very interesting that herbivory was the most important predictor variable. I would like to see a bit more about this in the discussion if possible.

Discussion:

Line 173: The phrase “structure variations” was unclear to me. Perhaps reword.
Line 175: What do you mean by “robustness” here. Do you mean ARISA is a good way to detect large differences in community composition (as suggested by the citation)?
Line 177: Please reword slightly. I understood this sentence to mean that the bacterial assemblage differed in leaves removed from the trees in spring and autumn. Is that correct?

Experimental design

The questions addressed by the paper are timely and important. Not much is known regarding how plant traits affect microbes, or how microbial communities shift over time. The use of ARISA precludes taxon-specific analyses, but still provides some interesting insights.

Validity of the findings

The experimental design was highly nested with multiple samples taken from the same branch, and multiple branches from the same tree. There was no mention of how this was accounted for statistically, or how temporal autocorrelation was addressed. The statistical methods should be improved.

I would also like to see a bit more in the discussion regarding why herbivory may be linked to shifts in community composition. This was a very interesting result, but was not discussed in detail.

---

## Round 0.2 · Minor Revisions

The reviewer and I concur that the manuscript has been much improved. However there are still some parts of the text that requires a carefull rephrasing (see reviewer's comments) in order to fit better with the actual results you are presenting.

·

Basic reporting

I commend the authors on adding a lot more useful content to their manuscript and reworking the statistical analyses. The paper is much improved. I have a few additional suggestions to consider. The most important suggestion is to be careful to not ascribe causality in cases where it cannot be determined due to the correlative nature of the data (see below).

Experimental design

Lines 175 onward: It is a bit unclear which set of leaves you sampled for herbivory? Did you look at the leaves that you sequenced or the leaves for which you measured traits, or some other set of leaves. Perhaps a sentence explaining this to the reader would be useful here.

Also, did you record each damage type as a categorical variable (i.e. either present or absent) or try to measure herbivory as a continuous variable (e.g. how much leaf was eaten)? I am guessing you used some continuous variable as you later mention that herbivory was a response in your linear mixed effects models. More detail here describing how you measured herbivory quantitatively and combined data from different types of herbivory would be helpful.

Line 226: The sentence, "Among the insect mediated damage types searched in our plots, the herbivory-mediated damage was due to the damage-types of hole feeding, margin feeding and sucking" is unclear to me. Do you mean that hole feeding, margin feeding, and sucking were the only types of herbivory observed?

Line 275: You say that herbivory was a significant association with shifts in endophyte assemblage over time as shown by CCA. However, it is unclear to me how "herbivory" was combined among the damage types observed? Was each damage type quantified in some way and these all added together? Why not analyze each damage type separately to gain additional insight?

Validity of the findings

The findings are interesting, but too much causality is ascribed in the wording of the results. The authors will want to rephrase sentences here and there to reflect the limitations due to the correlative nature of the data (see below).

Additional comments

Line 293 (and last line in abstract): Here you use the word "influenced" in relation to the association between stomata length and shifts in the endophyte assemblage. It would be better to be a bit more cautious here and perhaps use "associated" instead. Indeed, throughout the discussion it would be wise to make sure results are discussed as interesting correlations and avoid ascribing causality.

As another example, the last line of the abstract uses the word "drivers". Again, framing this result as an association or correlation would be more appropriate, as it is not possible to say if these traits cause the temporal shifts in the endophyte community or not. Any number of ecological contingencies shift from spring to autumn and would be associated with changes in the endophyte assemblage over that time period. Consequently, careful wording is necessary when discussing your interesting correlative results.

I suggest being more cautious in the conclusion statement as well. Your speculation that stomata and herbivory is important for endophyte colonization is well founded, and supported by your data, however much additional work will need to be undertaken before one can say what the "main drivers" of temporal change in endophyte communities are.

Even the title could probably be toned down. As an example: "Temporal shifts in endophyte bacterial community composition of sessile oak (Quercus petraea) are linked to foliar nitrogen, stomatal length, and herbivory" or something like that would be a bit more accurate given the correlative nature of the results.

---

## Round 0.3 · accepted · Accept

The reviewers and I agree that all the requested changes have been addressed and that your manuscript is now suitable for publication in PeerJ.

#